# Survival of *Escherichia coli* O157 in autoclaved and natural sandy soil mesocosms

Christopher A. Baker[1], Shinyoung Lee[2], Jaysankar De [1], Kwangcheol C. Jeong[2], Keith R. Schneider [1]*

1 Department of Food Science and Human Nutrition, University of Florida, Gainesville, FL, United States of America, 2 Department of Animal Sciences, Emerging Pathogens Institute, University of Florida, Gainesville, FL, United States of America

* keiths29@ufl.edu

**Data Availability Statement:** The datasets "Microbiota of autoclaved and natural sandy soil mesocosms" Accession: PRJNA589186 for this study can be found in NBCI Trace Archive NCBI

## Abstract

While the soil microbiome may influence pathogen survival, determining the major contributors that reduce pathogen survival is inconclusive. This research was performed to determine the survival of *E. coli* O157 in autoclaved and natural (unautoclaved) sandy soils. Soils were inoculated with three different *E. coli* O157 strains (*stx1+/stx2+*, *stx1-/stx2-*, and *stx1-/stx2+*), and enumerated until extinction at 30˚C. There was a significant difference in the survival of *E. coli* O157 based on soil treatment (autoclaved versus natural) at 30˚C on days 1 ($P = 0.00022$), 3, ($P = 2.53e-14$), 7 ($P = 5.59e-16$), 14 ($P = 1.072e-12$), 30 ($P = 7.18e-9$), and 56 ($P = 0.00029$), with greater survival in autoclaved soils. The time to extinction (two consecutive negative enrichments) for all three strains was 169 and 84 days for autoclaved and natural soils, respectively. A separate *E. coli* O157 trial supplemented with 16S rRNA gene sequencing of the soil microbiome was performed at 15˚C and 30˚C on days 0, 7, 14, and 28 for each soil treatment. Greater species richness (Chao1, $P = 2.2e-16$) and diversity (Shannon, $P = 2.2e-16$) was observed in natural soils in comparison with autoclaved soils. Weighted UniFrac (beta-diversity) showed a clear distinction between soil treatments ($P = 0.001$). The greatest reduction of *E. coli* O157 was observed in natural soils at 30˚C, and several bacterial taxa positively correlated (relative abundance) with time (day 0 to 28) in these soils ($P < 0.05$), suggesting that the presence of those bacteria might cause the reduction of *E. coli* O157. Taken together, a clear distinction in *E. coli* O157 survival, was observed between autoclaved and natural soils along with corresponding differences in microbial diversity in soil treatments. This research provides further insights into the bacterial taxa that may influence *E. coli* O157 in soils.

## Introduction

Contamination of fresh produce due to Shiga toxin-producing *Escherichia coli* (STEC) is a major concern for the produce industry. This has been highlighted by several high-profile *E. coli* O157:H7 outbreaks such as the spinach outbreak in 2006 with 199 illnesses, 102 hospitalizations, and three deaths from 26 states [1, 2]. Overall, produce-related foodborne outbreaks

Sequence Read Archive: https://www.ncbi.nlm.nih.gov/bioproject/PRJNA589186.

**Funding:** The author(s) received no specific funding for this work.

**Competing interests:** The authors have declared that no competing interests exist.

over the past several decades led to the implementation of the Produce Safety Rule (PSR) within the US Food Safety Modernization Act (FSMA), which was enacted in 2011 [3]. Even with the new regulation, the produce industry still struggles with outbreaks, which is exemplified by the two 2018 romaine lettuce outbreaks [4, 5] and the more recent 2019 romaine lettuce outbreak [6]. The PSR puts tremendous emphasis on water quality for both pre- and post-harvest use, while the use of soil amendments is somewhat undefined. Subpart F within the PSR focuses on biological soil amendments of animal origin (BSSAO), and the current harvest interval for untreated BSAAO is 'reserved' until a feasible and data-driven regulation can be set [7].

Many studies have focused on the survival of foodborne pathogens in manure-amended and non-amended soils [8–10]. Variation in survival duration can vary considerably depending on abiotic factors such as temperature, soil type, and soil structure [11–14]. In addition to abiotic factors, determining the influence of indigenous soil microbiota on pathogens contributes to the understanding of pathogen survival within soil environments prior to the harvest of produce.

It has been suggested that soils with a more diverse bacterial taxa profile are less likely to be colonized by a foreign microbe introduced into the soil [15]. This concept has been evaluated with a focus on foodborne pathogen establishment and survival in soils based on biodiversity [10, 16–23] although few studies have identified specific bacterial taxa within soils that may cause variation in pathogen survival [19]. Previous studies have investigated the influence of biotic factors on pathogen survival by altering and/or eliminating microbial populations in soil [17, 23] or compost [18] via steam sterilization (autoclaving) for comparison with natural (unautoclaved) soils. Greater survival of *E. coli* O157 has been observed in autoclaved soils when compared to natural soils [17, 20, 23, 24], which suggests that certain soil microorganisms that are reduced by heat treatments likely contribute to the reduction of *E. coli* O157 in soils.

This study was performed to further elucidate *E. coli* O157 survival in soil commonly used for agriculture in Florida, with an altered bacterial taxa profile following heat treatment. The survival of three *E. coli* O157 strains with different *stx* types was evaluated until extinction in autoclaved and natural sandy soil at 30˚C, and an additional trial implementing metagenomic analysis was performed to identify soil bacteria that may impact *E. coli* O157 survival in autoclaved and natural soil at 15 and 30˚C. This research provides an analysis of bacterial taxa that may contribute to *E. coli* O157 survival in sandy soil.

## Methods

### Soil and mesocosm preparation

Soil was collected from the University of Florida's Suwannee Valley Agricultural Experiment Station in Live Oak, FL in plastic-lined coolers, transported to the laboratory and stored at room temperature until further use. The soil site was selected based on previous studies investigating pathogen survival in soil [13] and to represent Floridian soils. Two soil treatments were performed to obtain autoclaved and natural soil as previously described [25]. Prior to autoclave treatment, soil was sieved (2 mm) and sterile spring water (Nestle Waters North America Inc., Stamford, CT) was added in 100 mL increments with subsequent mixing to obtain a 60% volumetric water content (VWC), measured with a GS3 soil moisture probe (Decagon Devices, Inc., Pullman, WA).

Soil was incubated at room temperature for 48 h to promote the growth of natural microbiota. Soil was autoclaved (121˚C, 15 PSI) for 1 h, and subsequently incubated at room temperature for 48 h to promote regrowth of microbial populations. This procedure was repeated,

and spring water was added as previously described to obtain a final 10% VWC. Soil was distributed into sterile sampling bags (7 x 12") (Fisher) to obtain 1 kg mesocosms. For natural soil mesocosms, soil was dried in a fume hood for 72 h and sieved (2 mm) to remove rocks and plant material. Water was added to soil in 100 mL increments and mixed thoroughly to obtain a VWC of 10% as previously described. Mesocosms, weighing 1 kg, were established in sterile sampling bags with autoclaved and natural soil. Each mesocosm was incubated at room temperature for 48 h to promote microbial growth prior to inoculation. A subsample of autoclaved and natural soil was analyzed for organic matter content (Walkley-Black titration), pH, macro- and micronutrients (Mehlich-3 extraction and analysis via inductively coupled plasma atomic emission spectroscopy (ICP-AES)) at the UF IFAS Extension Soil Laboratory (Gainesville, FL). Soil texture analysis by hydrometer method [26] was performed to determine percent clay, silt, and sand by the UF Soil and Water Sciences Department.

## Bacterial strains and inoculum preparation

The *E. coli* O157 strains used in this study have distinct *stx*-profiles (*stx1*+/*stx2*+ (6DL-17), *stx1*-/*stx2*- (5DOE-2), and *stx1*-/*stx2*+ (9OLM-10) and were obtained from bovine manure in Florida during previous research [27]. Cultures were streaked for isolation from cryovials onto tryptic soy agar (Difco, Sparks, MD) supplemented with 80 mg $L^{-1}$ rifampin (Thermo Fisher Scientific, Fair Lawn, NJ) (TSAR) and incubated at 37˚C for 24 h. An isolated colony was transferred to 10 mL of tryptic soy broth (Difco) supplemented with 80 mg $L^{-1}$ rifampin (TSBR) and incubated at 37˚C for 18 h, 100 rpm. One mL was transferred to 100 mL of TSBR, and incubated at 37˚C for 18 h, 100 rpm. After a final transfer and overnight growth as previously described, each culture was centrifuged at 1469 x g for 10 min. Supernatant was discarded, and cells were washed with 100 mL of 0.1% peptone water (PW) (Difco), vortexed for 10 s, and centrifuged as previously described. The cell wash step was repeated, and after the final centrifugation, cells were resuspended in 100 mL of 0.1% PW, vortexed for 10 s, and distributed into respective mesocosm bags in three consecutive 5 mL increments (15 mL total per mesocosm), with thorough mixing by hand after each inoculum addition for homogeneous distribution. One mL of each cell suspension was serially diluted in 9 mL of 0.1% PW, and the appropriate dilutions were plated onto TSAR and incubated for 24 h at 37˚C to determine the starting inoculum concentration in each mesocosm.

## *E. coli* O157 sampling until extinction in autoclaved and natural soil at 30˚C

The survival of *E. coli* O157 strains in autoclaved and natural soil was evaluated in triplicate for each strain and soil treatment at 30˚C (one strain per mesocosm, performed in triplicate for each soil treatment). On days 0, 1, 3, 7, 14, 30, 56, 84, 112, 139, and 169, 10 g of soil was transferred from each mesocosm into separate sterile sampling bags (7 x 12") (Fisher) containing 90 mL of 0.1% PW and stomached for 1 min. Stomached slurry was diluted and plated on TSAR and incubated at 37˚C for 24 h to determine the CFU $g^{-1}$ of soil in each sample. For control mesocosms, dilutions were plated onto trypic soy agar (TSA) and incubated at 37˚C for 24 h to determine the background microbiota in soils on days 0, 1, 3, 7, 14, 30, 56, and 84. When mesocosm samples approached the limit of detection (LOD) (1 $\log_{10}$ CFU $g^{-1}$ of soil), four 250 μL subsamples of stomached slurry was plated onto TSAR plates. If cell concentrations were below the LOD, a presence/absence detection method was performed– 10 mL of stomached slurry was added to 10 mL of double strength TSB supplemented with 160 mg $L^{-1}$ rifampin, incubated at 37˚C for 24 h, streaked onto TSAR, and incubated at 37˚C for 24 h. A loopful of colonies (if present) were added to a 1.5 mL centrifuge tube containing 100 μL of DNA

grade water (Fisher), heated at 100˚C for 20 min, centrifuged at 10,000 x g for 10 min, and supernatant was stored at -20˚C until further analysis. Quantitative PCR was performed to confirm the presence of *E. coli* O157 (*rfbE* target gene) as well as *stx* for respective mesocosm bags [25].

## 16S rRNA gene sequencing and analysis of microbial community in autoclaved and natural soil at 15 and 30˚C

Based on the results observed in the extinction trials, a supplementary fourth trial was performed incorporating metagenomic analysis of the autoclaved and natural soils on days 0, 7, 14, and 28. *E. coli* O157 survival in the autoclaved and natural soils was examined at both 15 and 30˚C on days 0, 3, 7, 14, 28, 56, 84, 112, and 140. Strain inoculum and soil treatments were prepared as previously described. Each strain (6DL-17, 5DOE-2, 9OLM-10) was separately inoculated into soil mesocosms for each soil treatment and incubation temperature (one trial). Mesocosm bags were sampled as previously described, and an additional 500 mg of soil was collected in a 1.5 mL microcentrifuge tube and stored at -80˚C until further analysis. Soil DNA was extracted from 250 mg of soil with a DNeasy PowerSoil Kit (Qiagen, Germantown, MD), adjusted to 10 ng/μL using DNA grade water (Fisher), and subjected to 16S rRNA gene PCR to amplify the V4 region of the 16S rRNA gene with dual-index primers [28]. The PCR amplicons were purified and normalized with SequalPrep Normalization plate kit (Invitrogen, USA). DNA concentration was measured with a Qubit 3.0 Fluorometer (Invitrogen, USA), and the same amount of DNA was pooled from each sample to make a DNA library. Quality of the pooled DNA library was confirmed with the Agilent 2200 TapeStation System and qPCR, and the final DNA library was sequenced using Illumina MiSeq platform with a 2 × 250 cycle cartridge at the University of Florida Interdisciplinary Center for Biotechnology Research.

Raw sequencing data was analyzed by QIIME pipeline (1.9.0) [29]. Raw sequencing reads were demultiplexed, assembled to paired-end reads, and chimeric sequences were filtered with a usearch61 method. The filtered sequences were assigned to an operational taxonomic unit (OTU) based on Silva database (https://www.arb-silva.de/documentation/release-132/) with 99% identity. The final OTU table was rarefied to the lowest sequencing depth (26,853), and one sample (natural microcosms at 30˚C on day 7) was excluded for further studies due to the lower number of reads.

The relative abundance of bacterial taxa between strains was averaged (n = 3) for each soil treatment at each temperature on days 0, 7, 14, and 28, except for natural microcosms at 30˚C on day 7 (n = 2). Unassigned taxa and taxa in low relative abundance (less than 2.5%) were grouped as "Other".

## Statistical analysis

One-way analysis of variance (ANOVA) was performed to compare the mean log CFU $g^{-1}$ on each sampling day based on strain type. A Student's two sample *t*-test was performed to compare differences in 1) mean log CFU $g^{-1}$ based on soil treatment ($P < 0.05$) and 2) mean log CFU $g^{-1}$ of uninoculated autoclaved and natural soils on each sampling day (days 0 to 84). Cell concentrations below the LOD of 1 log CFU $g^{-1}$ were assigned a value of 5 CFU $g^{-1}$ (0.69 log CFU $g^{-1}$) for positive enrichments, which is halfway between the LOD and 0 CFU $g^{-1}$, and 1 CFU $g^{-1}$ (0 log CFU $g^{-1}$) for negative enrichments. To compare dissimilarity of microbiota structures between treatments, an analysis of similarities (ANOSIM) was applied based on weighted UniFrac distances using QIIME. The alpha diversity (diversity within samples) was determined between treatments using Chao1 and Shannon diversity index [30]. Pearson correlations were performed on 30˚C natural soils to determine bacterial taxa that positively

correlated with time. Statistical analyses were performed using R version 3.4.3 (http://www.R-project.org). All statistics were performed at a significance level 0.05.

# Results

## Strains and soil

The strains used in this study were obtained from a bovine manure survey in Florida [25]. The soil (Candler sand) used in this study was obtained from the University of Florida's Suwannee Valley Agricultural Experiment Station (Live Oak, FL), which consisted of high levels of sand (97.52%) and low levels of clay (1.66%) and silt (0.83%). Similar pH (7.3), organic matter (0.50–0.57%), macro- and micronutrient levels were observed between autoclaved and natural soils, with the exception of manganese (Mn), which was more extractable in autoclaved soil (31.2 mg kg$^{-1}$) in comparison to natural soil (14.2 mg kg$^{-1}$).

## Indigenous soil microbiota in autoclaved and natural soil at 30˚C

Background microbial population levels for autoclaved and natural soils ranged from 5.63 and 7.59 log CFU g$^{-1}$ from days 0 to 84 for both soil treatments at 30˚C. There were minimal differences ($P > 0.05$) in culturable soil microbial populations over 84 days of incubation at 30˚C. There was a significant difference in indigenous soil microbiota between uninoculated autoclaved and natural soils only on days 1 ($P = 0.014$) and 3 ($P = 0.035$), with similar concentrations among treatments observed thereafter (Table 1).

## *E. coli* O157 survival based on soil treatment and strain

There was a significant difference in *E. coli* O157 mean log CFU g$^{-1}$ based on soil treatment at 30˚C on days 1 ($P = 0.00022$), 3 ($P = 2.53e\text{-}14$), 7 ($P = 5.59e\text{-}16$), 14 ($P = 1.072e\text{-}12$), 30 ($P = 7.18e\text{-}9$), and 56 ($P = 0.00029$) (Table 2). The first strain to reach the LOD (1 log CFU g$^{-1}$) was *E. coli* O157 *stx1+/stx2+* on day 30 in natural soil and *E. coli* O157 *stx1-/stx2+* on day 56 in autoclaved soils. All three strains were below the LOD on day 139 and 56 for autoclaved and natural soils at 30˚C, respectively. Total extinction (two consecutive negative enrichments, all three strains) was observed on day 169 and 84 for autoclaved and natural soils, respectively (Table 2). There was no significant difference ($P > 0.05$) in mean log CFU g$^{-1}$ of individual strains evaluated in both autoclaved and natural soil at 30˚C on any sampling day (Table 2).

**Table 1. Aerobic plate counts log CFU g$^{-1}$ in uninoculated autoclaved and natural soil mesocosms over 84 days of incubation at 30˚C.**

| Day | Log CFU g$^{-1}$ | |
|:---:|:---:|:---:|
| | Autoclaved Soil[a] | Natural Soil[a] |
| 0 | 7.39 ± 0.37a | 7.59 ± 0.04a |
| 1 | 7.49 ± 0.38a | 6.35 ± 0.27b |
| 3 | 6.65 ± 0.28a | 5.63 ± 0.49b |
| 7 | 7.30 ± 0.64a | 5.96 ± 0.12a |
| 14 | 7.08 ± 0.50a | 5.97 ± 0.12a |
| 30 | 6.78 ± 0.44a | 6.14 ± 0.33a |
| 56 | 6.73 ± 0.44a | 5.94 ± 0.14a |
| 84 | 6.78 ± 0.39a | 6.35 ± 0.22a |

[a]Means (n = 3) with the same letter in each row (a to b) are not significantly different ($P > 0.05$). Values represent log CFU g$^{-1}$ mean ± standard deviation

**Table 2. Comparison of *E. coli* O157 log CFU g$^{-1}$ at 30˚C until extinction.**

| Day | Autoclaved Soil[a] | | | Natural Soil[a] | | |
|---|---|---|---|---|---|---|
| | *stx1+/stx2+* | *stx1-/stx2-* | *stx1-/stx2+* | *stx1+/stx2+* | *stx1-/stx2-* | *stx1-/stx2+* |
| 0 | 7.22 ± 0.21a | 7.22 ± 0.12a | 7.22 ± 0.31a | 7.09 ± 0.27a | 6.97 ± 0.03a | 7.16 ± 0.08a |
| 1 | 7.48 ± 0.03a | 7.55 ± 0.13a | 7.58 ± 0.03a | 7.11 ± 0.21b | 6.91 ± 0.39b | 7.09 ± 0.37b |
| 3 | 7.37 ± 0.17a | 7.38 ± 0.05a | 7.40 ± 0.12a | 5.82 ± 0.08b | 6.01 ± 0.10b | 5.88 ± 0.18b |
| 7 | 7.24 ± 0.07a | 7.25 ± 0.03a | 7.29 ± 0.27a | 5.12 ± 0.14b | 5.33 ± 0.04b | 5.20 ± 0.12b |
| 14 | 7.07 ± 0.24a | 7.17 ± 0.37a | 6.47 ± 0.11a | 4.17 ± 0.12b | 4.15 ± 0.13b | 4.12 ± 0.15b |
| 30 | 5.77 ± 0.43a | 5.96 ± 0.90a | 4.58 ± 0.54a | 1.13 ± 0.51b | 1.26 ± 0.24b | 2.27 ± 0.14b |
| 56 | 1.89 ± 0.26a | 1.88 ± 0.76a | 0.43 ± 0.75a | 0.00 ± 0.00b | 0.00 ± 0.00b | 0.00 ± 0.00b |
| 84 | 0.00 ± 0.00 | 0.47 ± 0.40 | 0.00 ± 0.00 | 0.00 ± 0.00 | 0.00 ± 0.00 | 0.00 ± 0.00 |
| 112 | 0.23 ± 0.40 | 0.47 ± 0.40 | 0.00 | - | - | - |
| 139 | 0.00 | 0.00 ± 0.00 | - | - | - | - |
| 169 | 0.00 | 0.00 ± 0.00 | - | - | - | - |

[a]Values represent log CFU g$^{-1}$ mean ± standard deviation (n = 3). Values without a standard deviation are from samples with a single enrichment. Means (n = 3) with the same letter in each row (a to b) are not significantly different ($P > 0.05$). Statistical analyses were not performed after day 56 due to samples being below the LOD.

One trial was performed at 15 and 30˚C in autoclaved and natural soils for each *E. coli* O157 strain (metagenomics trial). The mean log CFU g$^{-1}$ (combined strains) was determined on each sampling day for comparison among soil treatments at each temperature (Table 3). Statistical analysis was not performed based on mean log CFU g$^{-1}$ in the metagenomics trail. However, similar mean log CFU g$^{-1}$ were observed between the extinction and metagenomics trials performed at 30˚C, and greater survival was observed at 15˚C in both soil treatments (Tables 2 and 3). On day 84, the mean log CFU g$^{-1}$ at 15˚C was 4.81 ± 0.18 in autoclaved soils and 2.99 ± 0.74 in natural soils compared to being below the LOD in both experiments at 30˚C (Tables 2 and 3). Although similar *E. coli* O157 concentrations were observed at 15 and 30˚C from day 0 to 28 in autoclaved soils, *E. coli* O157 concentrations declined until extinction (day 140) at 30˚C while remaining at 4.50 ± 1.57 log CFU g$^{-1}$ at 15˚C on day 140 (Table 3). Similarly, greater survival was observed in natural soils at 15˚C versus 30˚C–on day 28, *E. coli* O157 concentrations were at 4.95 ± 0.54 at 15˚C versus 1.29 ± 1.03 at 30˚C. On day 56, *E. coli* O157 concentrations were at 3.52 ± 0.44 at 15˚C versus 0.70 ± 0.00 log CFU g$^{-1}$ at 30˚C in natural soils (Table 3).

**Table 3. Comparison of *E. coli* O157 log CFU g$^{-1}$ (combined strains) in metagenomics trial.**

| Day | 30˚C Soil | | 15˚C Soil | |
|---|---|---|---|---|
| | Autoclaved[a] | Natural[a] | Autoclaved[a] | Natural[a] |
| 0 | 7.18 ± 0.32 | 6.58 ± 0.26 | 7.28 ± 0.25 | 7.06 ± 0.10 |
| 3 | 7.54 ± 0.28 | 5.71 ± 0.12 | 7.17 ± 0.35 | 7.09 ± 0.51 |
| 7 | 7.64 ± 0.29 | 5.37 ± 0.29 | 7.30 ± 0.25 | 6.24 ± 0.64 |
| 14 | 7.52 ± 0.34 | 3.98 ± 0.65 | 7.31 ± 0.17 | 5.69 ± 0.44 |
| 28 | 6.99 ± 0.11 | 1.29 ± 1.03 | 6.91 ± 0.44 | 4.95 ± 0.54 |
| 56 | 4.34 ± 1.17 | 0.70 ± 0.00 | 6.49 ± 0.67 | 3.52 ± 0.44 |
| 84 | 0.70 ± 0.00 | 0.00 ± 0.00 | 4.81 ± 0.18 | 2.99 ± 0.74 |
| 112 | 0.00 ± 0.00 | 0.00 ± 0.00 | 5.20 ± 1.04 | 1.93 ± 1.15 |
| 140 | 0.00 ± 0.00 | 0.00 ± 0.00 | 4.50 ± 1.57 | 1.39 ± 0.75 |

[a]Combined (mean ± standard deviation) log CFU g$^{-1}$ for each strain, one trial (n = 3)

### *E. coli* O157 population levels and soil metagenomic analysis

When comparing the bacterial distribution between the two soil treatments, there was a clear distinction between autoclaved and natural soil samples (Fig 1). While autoclaved soil samples consisted primarily of Proteobacteria and Firmicutes at both 15˚C and 30˚C on days 0, 7, 14, and 28 (Figs 2 and 3), there was a larger variation among autoclaved soil samples for weighted UniFrac distances ($P$ = 0.001) (Fig 1). At 30˚C, phyla in addition to Proteobacteria and Firmicutes were identified in autoclaved soil samples on days 7, 14, and 28, which was not observed at 15˚C except on day 28 with Actinobacteria (1.4%) and Bacteroidetes (0.1%) (Figs 2 and 3). In autoclaved soils at 30˚C, Verrucomicrobia increased from 4.2% on day 14 to 23.0% on day 28 as well as Actinobacteria from 1.4% to 3.9% to 5.0% on days 7, 14, and 28, respectively (Figs 2 and 3).

While there were minimal changes at the phylum level in autoclaved soils at 15˚C consisting primarily of Proteobacteria and Firmicutes, a more in-depth analysis at the class level revealed that within the Proteobacteria phylum, *Gammaproteobacteria* abundance dominated on day 0 (67%) followed by a lower relative abundance on days 7 (46.6%) 14 (52.0%), and 28 (40.0%) due to the increasing abundance of *Betaproteobacteria* on days 7 (26.1%), 14 (26.9%), and 28 (28.6%). *Gammaproteobacteria* abundance also decreased over time in autoclaved soils at 30˚C, while *Betaproteobacteria* abundance remained low at 5.1 to 5.6% abundance over days 7 to 28. *Alphaproteobacteria* abundance was less than 1% on days 7 and 14, and 3.9% on day 28 in autoclaved soil at 15˚C compared to 6.0% (day 7), 7.1% (day 14), and 13.2% (day 28) at 30˚C. Similarly, while absent at 15˚C, the relative abundance of *Clostridia* in autoclaved soils at 30˚C of 17.7%, 25.2%, and 13.0% was observed on days 7, 14, and 28, respectively (S1 Table).

In comparison to autoclaved soil samples, natural soil samples indicated similar bacteria composition among the samples (Fig 1). While natural soils exhibited fewer differences among

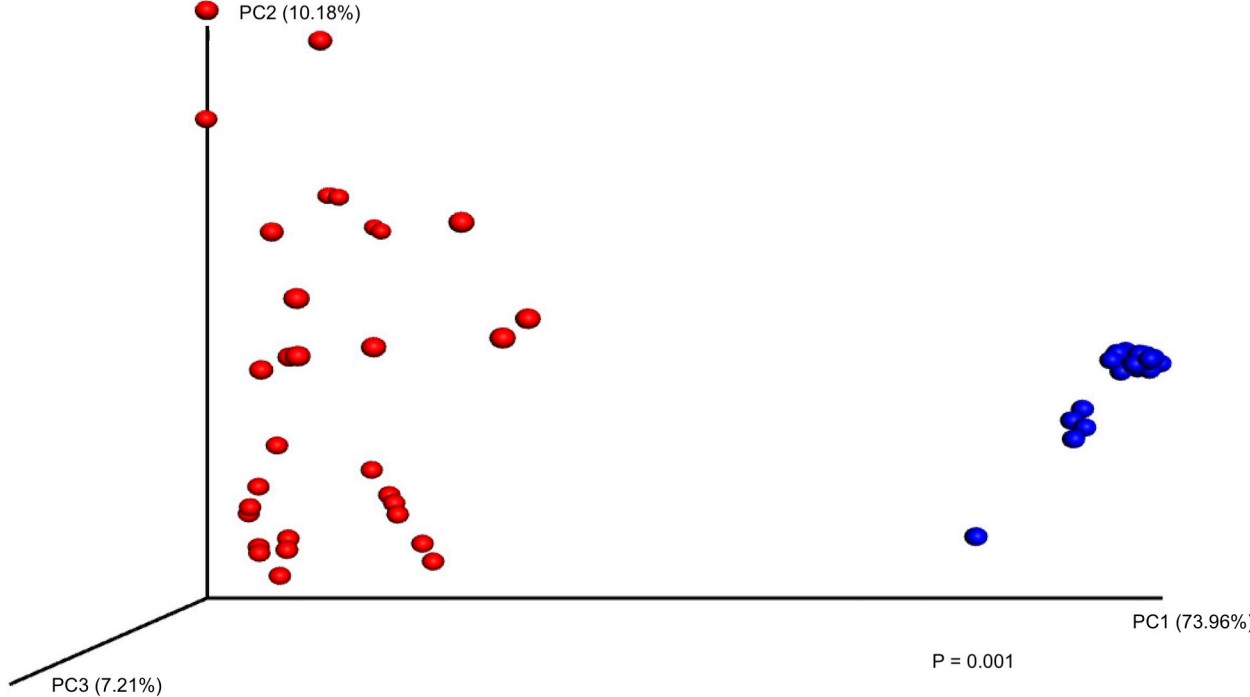

**Fig 1.** Principal Coordinate Analysis (PCoA) plot based on weighted UniFrac distances for autoclaved (red, n = 32) and natural (blue, n = 31) soil ($P$ = 0.001).

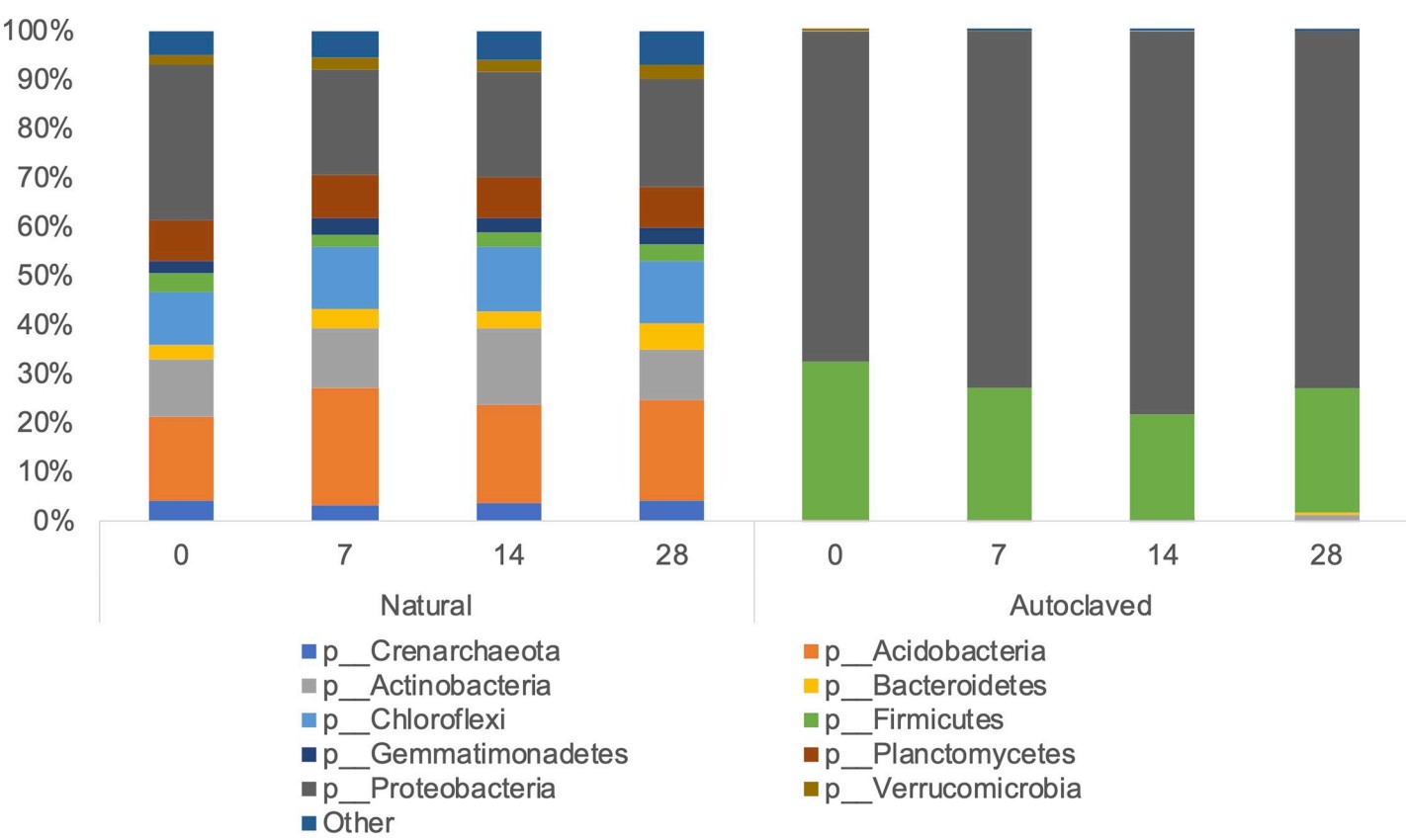

**Fig 2. The relative abundance of bacterial taxa (phylum) for natural (Nat) and autoclaved (Auto) soil at 15°C, days 0, 7, 14, 28.**

samples, many more bacterial taxa were represented in natural soils in comparison to autoclaved soils. Both Chao1 ($P$ = 2.2e-16), reflecting species richness, and Shannon indexes ($P$ = 2.2e-16), reflecting both species richness and evenness, were significantly different among autoclaved and natural soils (Table 4).

Natural soil samples at both 15°C and 30°C consisted primarily of following phyla: Proteobacteria (20.0–32.0%), Acidobacteria (17.0–24.2%), Actinobacteria (10.5–15.3%), Plantomycetes (8.1–9.0%), and Other (4.8–9.9%) (Figs 2 and 3). In natural soils on day 28, differences in *E. coli* O157 concentrations between 15°C (4.95 ± 0.54 log CFU g$^{-1}$) and 30°C (1.29 ± 1.03 log CFU g$^{-1}$) samples were observed (Table 3), with distinct microbiota structures between soil temperatures based on weighted UniFrac distances ($P$ = 0.001), suggesting microbial community of natural soils might influence *E. coli* O157 survival in soil (Fig 4).

To determine specific bacteria taxa that may contribute to *E. coli* O157 decline in soils, the taxa that positively correlated with time ($P < 0.05$) were identified in natural soils at 30°C (Fig 5) due to the greatest decline of *E. coli* O157 in these soils over time. Of identified bacterial taxa with a significantly positive correlation, only classified taxa with a relative abundance greater than 0.05% on day 28 were included (Fig 5). *Acidimicrobiales*, *Patulibacteraceae*, *Gaiellaceae*, *Conexibacteraceae*, *Pseudonocardia*, and *Pilimelia* were identified within the phylum Actinobacteria. Of these, the greatest relative abundance was *Gaiellacae*, which increased from 1.75 to 2.26% over 28 days. *Acidimicrobiales* increased from 0.43 to 0.78% over 28 days. Members of the Verrucomicrobia phylum that positively correlated with time were *Pedosphaerales*, *Verrucomicrobiaceae*, *Ellin517*, and *Opitutus*.

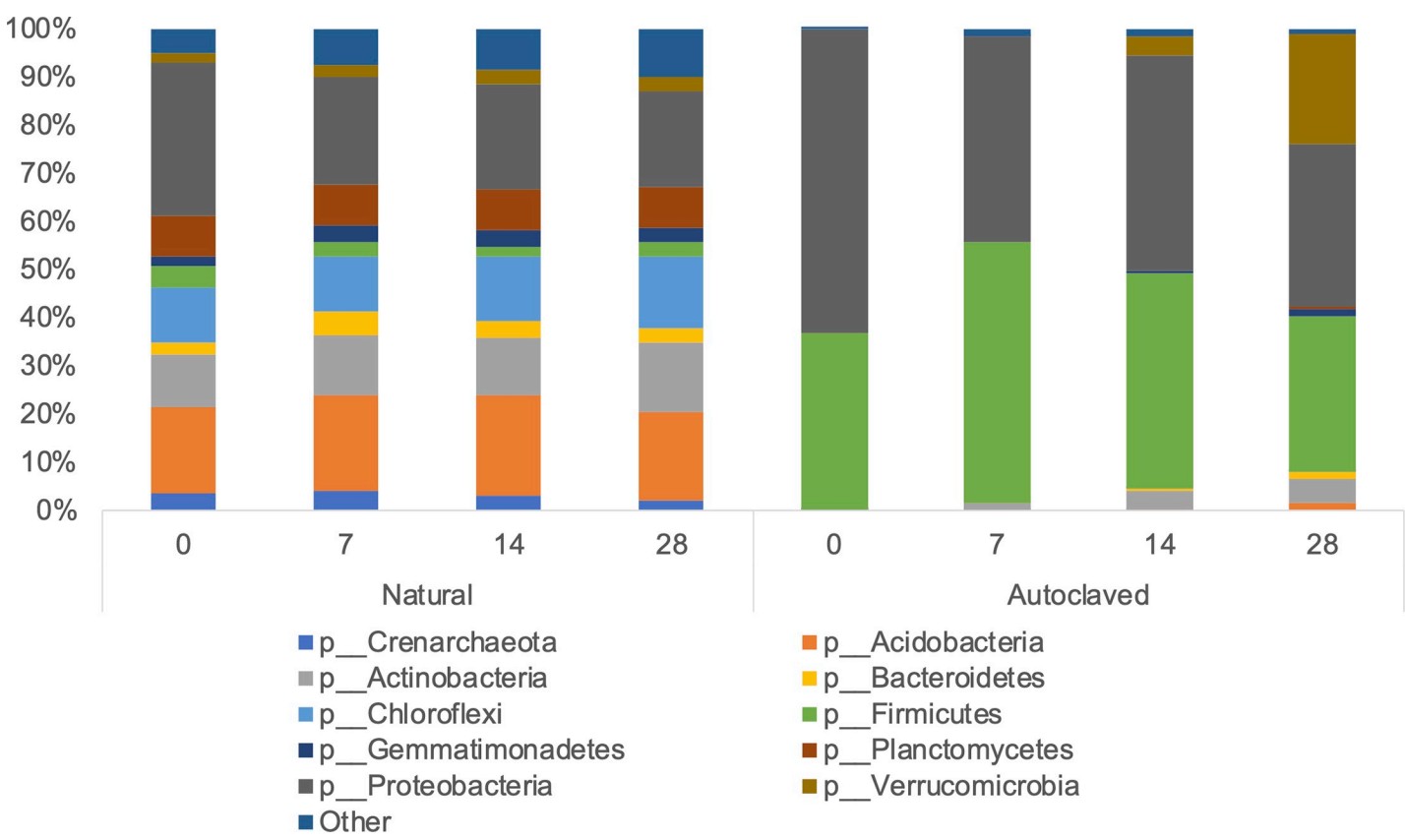

**Fig 3. The relative abundance of bacterial taxa (phylum) for natural (Nat) and autoclaved (Auto) soil at 30˚C, days 0, 7, 14, 28.**

Notable bacteria taxa within the Proteobacteria phylum include *Deltaproteobacteria*, *Myxococcales*, *Bacteriovoracaceae*, *Pedomicrobium*, and *Rhodoplanes*. *Gemmatimonadetes*, *Ellin6529*, *Phycisphaerales*, *Nitrospira*, and *Saprospiraceae* bacterial taxa were also identified to positively correlate with time ($P < 0.05$) (Fig 5).

## Discussion

This study was performed to identify the bacterial taxa that may be responsible for accelerated *E. coli* O157 reduction in sandy soil. The three *E. coli* O157 strains evaluated in this study possessed different *stx* profiles and were isolated from bovine manure on three geographically distinct farms in Florida during a previous study [25]. Although production of Stx has been hypothesized as a means of evading predation and enhancing survival [31], it is also possible that Stx production, especially in the soil environment, could lead to reduced survival based

**Table 4. Chao1 and Shannon diversity indices for autoclaved and natural soils.**

| Index | Treatment | | *P*-value |
|---|---|---|---|
| | Autoclaved[a] | Natural[a] | |
| Chao1 | 1285.96 ± 10.03 | 8567.55 ± 13.60 | 2.2e-16 |
| Shannon | 4.41 ± 0.05 | 10.58 ± 0.02 | 2.2e-16 |

[a]Values represent mean ± standard deviation of autoclaved and natural samples

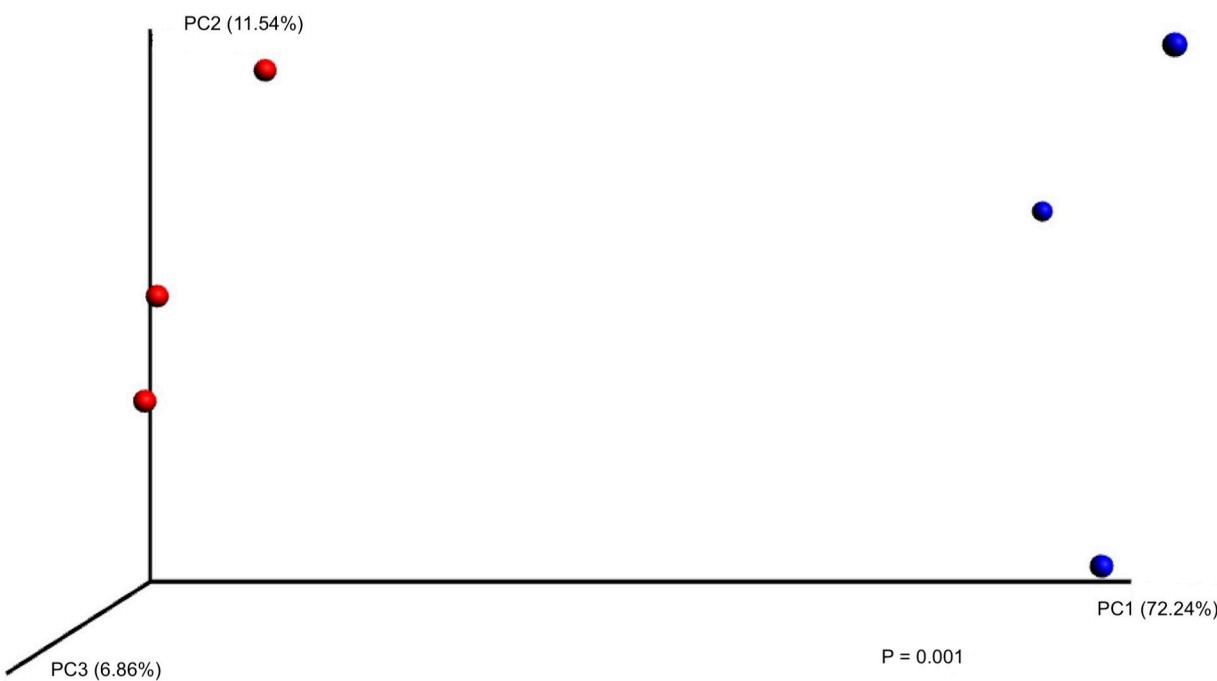

**Fig 4.** Principal Coordinate Analysis (PCoA) plot based on weighted UniFrac distances for 15˚C (red) and 30˚C (blue) natural soil on day 28 ($P = 0.001$).

on an increased energy expenditure [14]. Although it has been documented that an *stx- E. coli* O157 strain survived slightly better than an *stx+ E. coli* O157 strain at 23˚C [32], it is generally accepted that genotype related to STEC virulence factors do not impact survival in soils [8, 33], and similar findings were observed in this study.

With the exception of the micronutrient Mn, similar macro- and micronutrients were observed between the two soil treatments evaluated in this study. The higher level of extractable Mn observed in this study is similar to previous reports of increased soluble Mn concentrations following steam sterilization of soil [34, 35], though it is unclear whether the manganese levels observed in this study influence the survival of *E. coli* O157 in soils. Autoclaved soils might have resulted in altered organic carbon structures, although delineating physicochemical properties was not a focus of this study.

Minor variations in concentrations of indigenous soil microorganisms were observed over time at 30˚C in this study, which is congruent with previous evaluations of consistent indigenous soil microorganisms concentrations in unautoclaved soils via aerobic plate counts over time [13, 23]. Tanaka et al. [36] observed miminal changes in soil microbial cell concentration following steam sterilization in soils. These authors observed an initial soil bacterial population of 7.81 log CFU $g^{-1}$, and after autoclaving treatment (60˚C (soil temperature) for approximately 3 h), soil bacterial populations ranged from 7.13 to 8.39 log CFU $g^{-1}$ over a 12 day incubation period.

As noted by Kim et al. [18] and exemplified in this research, steam sterilization can be used to alter microbiota populations in compost and soil. Kim et al. [18] autoclaved compost for various lengths of time and temperatures to obtain approximate concentrations of indigenous mesophilic microorganisms within compost for subsequent analysis of *E. coli* O157 survival. By subjecting compost to 121˚C for 20 min (three times on three consecutive days), the authors observed an initital indigenous mesophilic population of approximately 2 log CFU $g^{-1}$

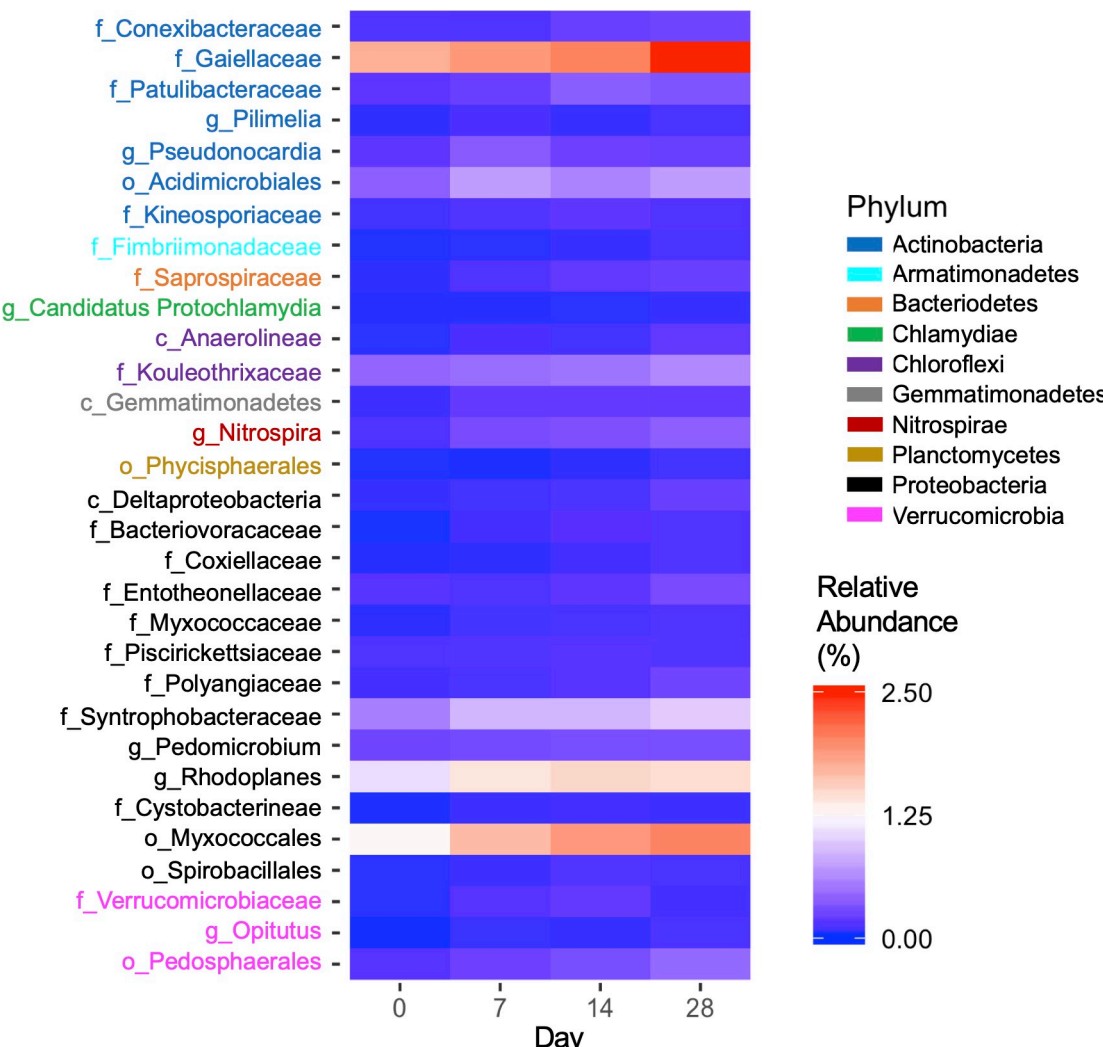

**Fig 5. Bacterial taxa in natural soil at 30˚C that positively correlated with time ($P < 0.05$).** Pearson correlations between relative abundance of bacterial taxa and time (day) were analyzed, and only classified taxa with a relative abundance above 0.05% on day 28 with a significant and positive correlation were included in the heatmap.

in the treated compost. They subsequently incubated the treated compost at 22˚C for 18, 24, 30, and 48 h, resulting in approximate mesophilic microorganism populations of 5.0, 6.0, 6.5 and 8.0 log CFU $g^{-1}$ [18]. In this study, similar aerobic plate counts were observed in both autoclaved and natural soils, which is likely a result of the proliferation of a few heat-resistant bacterial taxa due to substrate addition from decomposed microbes and/or reduced competition within the autoclaved soil [36].

While the aerobic plate counts were similar among uninoculated autoclaved and natural soils, *E. coli* O157 strains survived longer in autoclaved versus natural soils at 30˚C. Total extinction (two consecutive negative enrichments for each strain) was observed on day 169 and day 84 for autoclaved and natural soils, respectively, in the extinction trials. In the metagenomics trial, *E. coli* O157 declined to similar concentrations at each sampling point in comparison to the extinction trial at 30˚C. The most drastic decline in *E. coli* O157 was observed between day 56 and 84 in the metagenomics trial and between day 30 and 56 in the extinction

trial at 30˚C. Additionally, greater survival was observed at 15 versus 30˚C in both soil treatments.

Ibekwe et al. [24] compared the survival of *E. coli* O157:H7 in silty clay and loamy sand soils that were either autoclaved (1 h, cooled 24 h, repeat autoclaving 1 h) or unautoclaved and subsequently incubated at 20˚C. These authors observed approximately less than 3 log CFU $g^{-1}$ reduction in autoclaved loamy sand in comparison to over a 5 log CFU $g^{-1}$ reduction in unautoclaved loamy sand over 60 days at 20˚C. In silty clay soils, minor differences in *E. coli* O157: H7 populations were observed in both soil treatments during the first 30 to 40 days of incubation after which greater survival was observed in the autoclaved silty clay [24].

Moynihan et al. [20] incubated sandy loam and clay loam soils (both autoclaved (121˚C, 1 h twice over two days) and unautoclaved) initially at 4˚C for 6 days and observed minimal changes in non-toxigenic *E. coli* O157 populations. However, once soils were transferred to an 18˚C incubation to mimic warmer temperatures associated with Ireland spring conditions, a steady decrease in unautoclaved soil and slight increase in autoclaved soil of *E. coli* O157 populations was observed for 64 days of incubation, and these differences were attributed to an increased metabolic activity of the indigenous soil microbial community [20]. Similar conclusions were made by Jiang et al. [17] when observing greater survival of *E. coli* O157:H7 at 15 and 21˚C in manure-amended autoclaved sandy loam soils (121˚C for 20 min, three consecutive days) in comparison to manure-amended unautoclaved sandy loam soils when soils were amended with various ratios of manure:soil of 1:10, 1:25, 1:50, 1:100. These authors noted longer survival times in autoclaved soils amended with less manure (1:25, 1:50, 1:100) in comparison to the 1:10 manure:autoclaved soil at both temperatures [17]. While a direct comparison was not made between autoclaved and unautoclaved soils, Vidovic et al. [23] observed better survival of *E. coli* O157:H7 in autoclaved silty clay loam microcosms in comparison to manure-amended unautoclaved silty clay loam at -21, 4, and 20˚C.

van Overbeek et al. [22] implemented denaturing gradient gel electrophoresis (DGGE) analysis to evaluate the survival of *E. coli* O157 in 36 soils (sand and loam) amended with manure at 16˚C. Significant differences in irregularity were observed between two clusters based on Pearson correlations of 16S rRNA fingerprints as well as a positive correlation between irregularity and diversity for a subset of soil. It was suggested that an increase in diversity may contribute to irregularity, which is defined as how well a population decline follows a specific pattern (i.e., a high irregularity results in a lower accuracy of predicting population behavior) [22]. However, it should be noted that the time to reach the detection limit was not significantly different between these soils [22].

The diversity of bacterial populations in natural soil samples was much higher when compared among autoclaved soil samples based on the number of OTUs during sequencing for each sample. This was as expected as thermally-susceptible bacterial taxa were expected to be removed during autoclaving, but remained unaffected for the unautoclaved natural soil samples. While there was a greater diversity of taxa in natural soil samples, differences in the relative abundance of bacteria over time and temperature were lower in comparison to autoclaved soils. The larger diversity among autoclaved soil samples during the later stage of the experiment was likely due to the regrowth of bacterial taxa over time, especially at 30˚C. Future research should focus on extending the regrowth period (more than 48 h) after autoclaving soil prior to inoculation, which may allow the bacterial population to reach equilibrium in the soil. Additionally, shifts in the microbial diversity from soil mixing during sampling may occur and warrants further investigation.

Soil bacterial diversity in other regions and soil types can vary–sand and loam soils (n = 36) in the Netherlands clustered into three groups based on microbial diversity, with Shannon diversity indices ranging from 3.35 to 3.78 [22], while clay soils in France have exhibited a

Shannon diversity of 8.57 [37]. A higher bacterial diversity was observed in the Florida sandy soil examined in this study, although these comparisons should be carefully considered based on different sequencing approaches [38]. Ma et al. [19] did not observe an influence of bacterial diversity on *E. coli* O157 survival times across 32 soils with a wide range of soil texture properties from California and Arizona, and Shannon diversity indices ranges from 6.04 to 7.29. However, correlations between higher survival in the presence of Actinobacteria and Acidobacteria and suppression of survival in the presence of Proteobacteria and Bacteroidetes were noted [19]. In this study, nearly half (14/31) of the bacterial taxa that positively correlated with time ($P < 0.05$) in natural soils at 30°C were in the Proteobacteria phylum, which suggests that bacteria in the Proteobacteria phylum may suppress *E. coli* O157 survival in soil similar to previous work [19]. Among the Proteobacteria that positively correlated with time in natural soil at 30°C, *Myxococcales* and *Bacteriovoracaceae* have been associated with antagonistic activity against *E. coli*. *Myxococcales* include *Myxococcus xanthus*, which produce bioactive secondary metabolites and have been associated with predation of bacteria [39, 40]. Likewise, the family of *Bacteriovoraceae* include *Bdellovibrio* which are well characterized predators [41], and lysis of *E. coli* and other Gram-negative bacterial cells has been demonstrated [42, 43]. *Rhodoplanes* also increased in relative abundance over 28 days, which has previously been observed following the addition of a derivative *E. coli* O157 strain to two sandy soils [44].

Several taxa in Actinobacteria phylum positively correlated with time, with the greatest relative abundance of *Gaiellaceae*. The *Gaiellaceae* family is comprised of single Gram-negative species *Gaiella occulta* which was first isolated from a deep mineral water aquifer [45], and while further characterization of this organism progresses [46], knowledge of potential antimicrobial capabilities is limited. *Pseudonocardia* has been associated with antimicrobial properties, with a particular strain producing a carbohydrate-based compound effective against *Staphylococcus aureus* that withstands autoclaving and boiling treatment [47]. While there were low relative abundances of *Verrucomicrobiaceae*, primer bias may have led to a misrepresentation of the relative abundance, and these taxa may play an important role in soil microbial communities [48].

Temperature is an important abiotic factor that has shown to influence the death rates of pathogens in soil, with higher temperatures leading to more rapid die-off [11–13]. While higher temperatures may lead to a higher metabolic activity of *E. coli* O157 and more rapid utilization of nutrients and in the environment, similar effects could occur with indigenous soil microorganisms based on temperature [17, 23]. In natural soils, *E. coli* O157 decreased over 5 log CFU g$^{-1}$ by day 30 at 30°C, which could be due the higher bacteria taxa and/or an increased metabolic function as previously stated. In the current experiments, similar *E. coli* O157 concentrations were observed in autoclaved soils at both temperatures, which was not observed in natural soils. This data supports that temperature influences microbial activity and subsequent *E. coli* O157 survival, and may be accelerated in more diverse soils. Metagenomic data in this study shows that bacterial diversity increases over time in autoclaved soils, which may explain the increased decline in *E. coli* O157 at 30°C from day 56 to 84 at 30°C. While though no significant differences were found in the survival of the three strains at 30°C, a different strain-dependent impact of the lower temperature and/or on the detected taxa cannot be excluded and warrants further study.

Overall, this study provides an evaluation of representative bacterial populations in autoclaved and natural sandy soil at two temperatures, and their effect on *E. coli* O157 survival. Future experiments are needed to evaluate the impact of specific bacterial taxa on *E. coli* O157 survival in soils. It should be noted that other indigenous microbiota such as soil protists, fungi, and nematodes were likely removed while autoclaving soils, which could also influence pathogen survival. Additionally, farming practices that reduce soil bacterial diversity and any

associated risks should be considered [15, 37]. This data, along with metagenomic analyses of soil from other geographic regions, will be useful in determining the role of natural soil microbiome on the fate of *E. coli* O157 in soils.

## Supporting information

**S1 Table.**
(XLSX)

## Acknowledgments

The authors thank Chris R. Pabst and Bruna Bertoldi for technical assistance.

## Author Contributions

**Conceptualization:** Christopher A. Baker, Jaysankar De, Kwangcheol C. Jeong.

**Data curation:** Christopher A. Baker, Shinyoung Lee.

**Formal analysis:** Christopher A. Baker, Shinyoung Lee.

**Funding acquisition:** Keith R. Schneider.

**Investigation:** Christopher A. Baker.

**Methodology:** Christopher A. Baker, Kwangcheol C. Jeong, Keith R. Schneider.

**Project administration:** Keith R. Schneider.

**Supervision:** Kwangcheol C. Jeong, Keith R. Schneider.

**Writing – original draft:** Christopher A. Baker.

**Writing – review & editing:** Jaysankar De, Kwangcheol C. Jeong, Keith R. Schneider.

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
