## [Decision Letter · Decision Letter 0]

13 Apr 2020

PONE-D-20-03610

Survival of Escherichia coli O157 in Autoclaved and Natural Sandy Soil Mesocosms

PLOS ONE

Dear Dr.  Keith R. Schneider,

Thank you for submitting your manuscript to PLOS ONE. After careful consideration, we feel that it has merit but does not fully meet PLOS ONE’s publication criteria as it currently stands. Therefore, we invite you to submit a revised version of the manuscript that addresses the points raised during the review process.

We would appreciate receiving your revised manuscript by 30-05-2020. To enhance the reproducibility of your results, we recommend that if applicable you deposit your laboratory protocols in protocols.io, where a protocol can be assigned its own identifier (DOI) such that it can be cited independently in the future. For instructions see: http://journals.plos.org/plosone/s/submission-guidelines#loc-laboratory-protocols

We look forward to receiving your revised manuscript.

Kind regards,

Luigimaria Borruso

Academic Editor

PLOS ONE

Journal Requirements:

Reviewers' comments:

Reviewer's Responses to Questions

**Comments to the Author**

1. Is the manuscript technically sound, and do the data support the conclusions?

Reviewer #1: Yes

Reviewer #2: Partly

2. Has the statistical analysis been performed appropriately and rigorously? 

Reviewer #1: Yes

Reviewer #2: No

3. Have the authors made all data underlying the findings in their manuscript fully available?

Reviewer #1: Yes

Reviewer #2: Yes

4. Is the manuscript presented in an intelligible fashion and written in standard English?

Reviewer #1: Yes

Reviewer #2: Yes

5. Review Comments to the Author

Reviewer #1: In the manuscript titled: “Survival of Escherichia coli O157 in Autoclaved and Natural Sandy Soil Mesocosms”, the authors observe and compare survival dynamics of three strains (stx types) of E. coli O157 in native and autoclaved sandy soils. Further experiments were performed to compare survival at two temperatures (15°C and 30°C) and to investigate the microbial diversity within the soil samples over time using a high-throughput 16S sequencing approach. Ultimately, they observe that E. coli O157 survives better in a less diverse environment (autoclaved soil) and at cooler temperatures, likely due to reduced competition.

The article presents a valuable and detailed description of the soil-borne bacteria detected and supporting evidence for how these populations may influence the survival of E. coli O157 in native soils. The findings augment current knowledge on the topic of E. coli survival in soils, and provide a list of bacterial taxa that warrant further investigation. The article is well written and supported in the literature with references to 47 sources.

Major Points:

1. While there is a great discussion on the specific taxa that were observed in this study on sandy soil, there is little discussion on the overall microbial diversity. Is it know if the microbial composition observed is similar or varies from what has been seen in other soil types, geographies, agriculture or other samples?

Minor Points:

• Introduction – the current research is introduced in the context of produce safety but there is no discussion of the choice to focus on sandy soil. Is this a soil composition that is typical for agriculture in Florida, or for romaine lettuce, or simply what the authors had access to?

• Methods Section – suggest to reorder the methods with soil preparation first then bacterial strains and inoculum preparation, for simplicity for the reader.

• Lines 281-293 – referencing data presented in Supplemental Table? If so please add reference to that.

• Lines 320-322 – Suggest to break this sentence and slightly reconstructing it for simplicity: “… were identified. Of these, the greatest relative abundance was Gailellacae, which increased…. “

• Lines 355-357 – It has long been known that high levels of manganese can be toxic to E. coli (Silver S et al. 1972, Hantke K 1987), please compare the concentrations observed in the autoclaved soil (31.2 mg/kg) to previously described inhibitory concentrations of Mn.

• Line 415 –write out DGGE abbreviation

• Table 2

o Please add “(n = 3)” behind “mean” in c footnote

o Footnotes c and d should come first and do not require a letter, otherwise the letter should be somewhere appropriately located in the table itself.

• Tables 1, 3 and 4 – Similarly for the footnotes about explaining the values, either place a corresponding letter next to “autoclaved” and “natural” or omit the letter).

• Figures – are generally low quality and resolution should be improved.

o Text in figures 1 & 4 should be increased and spaced away from overlapping with axes

o Figure 2 & 3, suggest to use bars to span all natural and autoclaved samples, so that sample names can be simplified to days (temp is not needed for each sample name). I would also suggest to format the legend to classify based on kingdom, so that only phylum needs to be displayed for each entry, this would simplify interpretation for the reader.

o Figure 5 – suggest to add a heading above the phylum legend and remove the p_ in the naming.

• Supplemental File 1 (Table) –

o please include a note for interpretation of the sample ID code. Which I assume is autoclaved or natural – time point (days) – incubation temp – unknown number

o numeric values represent the proportion of the total population (1.0) with a given taxonomy by 16S sequencing analysis. – this should be noted somewhere.

Reviewer #2: The work of Baker et al., aims to determine the survival of three strains of E. coli O157 with different stx types in autoclaved and natural sandy soil at 30°C. A metagenomic analysis in autoclaved and natural soil at 15 and 30ºC was also performed, providing information on the bacterial taxa that may contribute to E. coli O157 survival in sandy soil.

In general, the work is well-written, the topic is interesting and it open stimulating perspectives to evaluate the impact of specific bacterial taxa on E. coli O157 survival in soils.

However, in my opinion, some methodological issues should be clarified better.

The authors should clearly explain the reason why they investigated E. coli survival and soil taxa just at 15°C in the metagenomic trial. Moreover, I don’t well understand if the metagenomic analysis have been performed in one sample co-inoculated with the three E. coli strains, or in three samples each inoculated with one E. coli strain. This information must be clearly provided in materials and methods. Anyway, though no significant differences were found in the survival of the three strains at 30°C, a different strain-dependent impact of the lower temperature and of (or on) the detected taxa cannot be excluded. Moreover, statistical analyses were not performed for the trial conducted at 15 and 30ºC in autoclaved and natural soils for metagenomic analysis, thus reducing the robustness of the results. The authors are invited to clarify these points.

Minor revisions

L.30. different

L93. Please, use a code to provide an identification at a strain level of the E. coli used in this work.

L133-135. Provide more details about the analysis of organic compounds, macro- and micronutrients

L142. 169 or 160?

L147. Why the background microbiota in soils was carried out on days 0, 1, 3, 7, 14, 30, 56, and 84, and not only until 160 days as for E. coli determination?

L165. mesocosm

L240-243. Please, rephrase the sentence

L251-253. This information is provided in a confusing way

L253-256. A similar difference (about 3-Log) is detectable at day 28

L346-348. If no significant differences were found in the survival, the sentence is a no-sense

L385-387. Rephrase the sentence in a more clear way

6. PLOS authors have the option to publish the peer review history of their article (what does this mean?). If published, this will include your full peer review and any attached files.

Reviewer #1: No

Reviewer #2: No

---

## [Author Response · Author response to Decision Letter 0]

25 Apr 2020

Reviewer #1:

Major Points:

While there is a great discussion on the specific taxa that were observed in this study on sandy soil, there is little discussion on the overall microbial diversity.

Is it know if the microbial composition observed is similar or varies from what has been seen in other soil types, geographies, agriculture or other samples?

Response: As recommended by Reviewer 1, we have added a discussion on potential differences in bacterial diversity between regions and soil types:

Before: Additionally, shifts in the microbial diversity from soil mixing during sampling may occur and warrants further investigation.

Ma et al. [19] did not observe an influence of bacterial diversity on E. coli O157 survival times across 32 soils from California and Arizona. However, correlations between higher survival in the presence of Actinobacteria and Acidobacteria and suppression of survival in the presence of Proteobacteria and Bacteroidetes were noted (19).

After: 

Additionally, shifts in the microbial diversity from soil mixing during sampling may occur and warrants further investigation.

Soil bacterial diversity in other regions and soil types can vary – sand and loam soils (n = 36) in the Netherlands clustered into three groups based on microbial diversity, with Shannon diversity indices ranging from 3.35 to 3.78 (22), while clay soils in France have exhibited a Shannon diversity of 8.57 (37). A higher bacterial diversity was observed in the Florida sandy soil examined in this study, although these comparisons should be carefully considered based on different sequencing approaches (38). Ma et al. [19] did not observe an influence of bacterial diversity on E. coli O157 survival times across 32 soils with a wide range of soil texture properties from California and Arizona, and Shannon diversity indices ranges from 6.04 to 7.29. However, correlations between higher survival in the presence of Actinobacteria and Acidobacteria and suppression of survival in the presence of Proteobacteria and Bacteroidetes were noted (19).

Minor Points:

Introduction – the current research is introduced in the context of produce safety but there is no discussion of the choice to focus on sandy soil. Is this a soil composition that is typical for agriculture in Florida, or for romaine lettuce, or simply what the authors had access to?

Although mentioned in the methods section, we have added a clearer explanation for the soil used in this study in the first sentence of the last paragraph in the introduction:

“This study was performed to further elucidate E. coli O157 survival in soil commonly used for agriculture in Florida, with an altered bacterial taxa profile following heat treatment.”

Methods Section – suggest to reorder the methods with soil preparation first then bacterial strains and inoculum preparation, for simplicity for the reader.

Response: As suggested by Reviewer 1, the soil preparation methods section has been moved to the front of the Methods section to prevent confusion in the Bacterial strain and inoculation preparation section.

Referencing data presented in Supplemental Table? If so please add reference to that.

Response: Yes. As suggested by Reviewer 1, we have added “(Supplemental Table 1)” to the end of this text.

Suggest to break this sentence and slightly reconstructing it for simplicity: “… were identified. Of these, the greatest relative abundance was Gailellacae, which increased…. “

Response: This sentence was split and reconstructed as suggested by Reviewer 1.

Before: “Within the phylum Actinobacteria, Acidimicrobiales, Patulibacteraceae, Gaiellaceae, Conexibacteraceae, Pseudonocardia, and Pilimelia were identified, with the greatest relative abundance of Gaiellacae increasing from 1.75 to 2.26% over 28 days. Acidimicrobiales increased from 0.43 to 0.78% over 28 days.”

After: Acidimicrobiales, Patulibacteraceae, Gaiellaceae, Conexibacteraceae, Pseudonocardia, and Pilimelia were identified within the phylum Actinobacteria. Of these, the greatest relative abundance was Gaiellacae, which increased from 1.75 to 2.26% over 28 days.

It has long been known that high levels of manganese can be toxic to E. coli (Silver S et al. 1972, Hantke K 1987), please compare the concentrations observed in the autoclaved soil (31.2 mg/kg) to previously described inhibitory concentrations of Mn.

Response: Based on Silver et al. (1972), we have calculated an inhibitory concentration of 549 mg/L in broth culture at 37ºC. Based on the value observed in soils in this study, we do not think it is necessary to include this data or discussion point as it may lead to confusion to the reader.

Write out DGGE abbreviation

Response: We have spelled out DGGE as suggested by Reviewer 1.

Table 2

Please add “(n = 3)” behind “mean” in c footnote

Response: This has been added as suggested by Reviewer 1.

Footnotes c and d should come first and do not require a letter, otherwise the letter should be somewhere appropriately located in the table itself.

Response: Footnotes b, c and d have been moved within footnote as suggested by Reviewer 1.

Tables 1, 3 and 4 – Similarly for the footnotes about explaining the values, either place a corresponding letter next to “autoclaved” and “natural” or omit the letter).

Response: For tables 1, 3, and 4, superscripts for footnote a have been added

Figures – are generally low quality and resolution should be improved.

Text in figures 1 & 4 should be increased and spaced away from overlapping with axes

Response: As suggested by Reviewer 1, the text in figures 1 and 4 has been increased and spaced away from axes. Unfortunately, resolution cannot be increased for figures 1 and 4 since they were generated with online software QIIME. We have improved the resolution of Figures 2-5 the best we can a minimum dpi of 300 as required by PLoS ONE.

Figure 2 & 3, suggest to use bars to span all natural and autoclaved samples, so that sample names can be simplified to days (temp is not needed for each sample name). I would also suggest to format the legend to classify based on kingdom, so that only phylum needs to be displayed for each entry, this would simplify interpretation for the reader.

Response: Figures 2 & 3 have been updated to remove temp, and spanning natural and autoclave sample labels, along with removing the kingdom classification.

Figure 5 – suggest to add a heading above the phylum legend and remove the p_ in the naming.

Response: A heading has been added and the p_ has been removed in Figure 5 as suggested by Reviewer 1.

Supplemental File 1 (Table) –

Please include a note for interpretation of the sample ID code. Which I assume is autoclaved or natural – time point (days) – incubation temp – unknown number

Numeric values represent the proportion of the total population (1.0) with a given taxonomy by 16S sequencing analysis. – this should be noted somewhere.

Response: As suggested by Reviewer 1, an explanation of Sample ID codes and relative abundance of total population has been added to Supplemental file 1 (see below):

Sample ID code:

“Treatment_Day_Temperature_Strain; 

"Auto_0_30_1" = Autoclaved_Day0_30ºC_Strain 1

Strain ID: 1 = 6DL-17; 5 = 5DOE-2; 9 = 9OLM-10; NC = no E. coli O157 added

Numeric values represent proportion of the total population (1.0) with a given taxonomy by 16S sequencing analysis”

Reviewer #2: The work of Baker et al., aims to determine the survival of three strains of E. coli O157 with different stx types in autoclaved and natural sandy soil at 30°C. A metagenomic analysis in autoclaved and natural soil at 15 and 30ºC was also performed, providing information on the bacterial taxa that may contribute to E. coli O157 survival in sandy soil. In general, the work is well-written, the topic is interesting and it open stimulating perspectives to evaluate the impact of specific bacterial taxa on E. coli O157 survival in soils.

However, in my opinion, some methodological issues should be clarified better.

The authors should clearly explain the reason why they investigated E. coli survival and soil taxa just at 15°C in the metagenomic trial.

Response: We are unsure of the misunderstanding that metagenomic analysis was only performed at 15ºC. Analysis was performed at both temperatures (15ºC and 30ºC) as stated in the text: “A metagenomic analysis in autoclaved and natural soil at 15 and 30ºC was also performed, providing information on the bacterial taxa that may contribute to E. coli O157 survival in sandy soil."

Moreover, I don’t well understand if the metagenomic analysis have been performed in one sample co-inoculated with the three E. coli strains, or in three samples each inoculated with one E. coli strain. This information must be clearly provided in materials and methods.

Response: We have updated the text in the Methods section to make it clearer on how soils were inoculated for the extinction trials and metagenomic trail.

“The survival of E. coli O157 strains in autoclaved and natural soil was evaluated in triplicate for each strain and soil type at 30ºC (one strain per mesocosm, performed in triplicate for each soil treatment).”

Updated section explaining the metagenomics trail:

“Each strain (6DL-17, 5DOE-2, 9OLM-10) was separately inoculated into soil mesocosms for each soil treatment and incubation temperature (one trial).”

Anyway, though no significant differences were found in the survival of the three strains at 30°C, a different strain-dependent impact of the lower temperature and of (or on) the detected taxa cannot be excluded.

Response: A statement to this point has been added to the discussion section.

Moreover, statistical analyses were not performed for the trial conducted at 15 and 30ºC in autoclaved and natural soils for metagenomic analysis, thus reducing the robustness of the results. The authors are invited to clarify these points.

Response: While statistical analysis were not performed in the single metagenomics trials, we do mention that similar E. coli O157 levels are present at 30ºC (in the metagenomics trial) for both soil treatments when comparing it to the triplicate extinction trial at 30ºC:

“Statistical analysis was not performed based on mean log CFU g-1 in the metagenomics trail. However, similar mean log CFU g-1 were observed between the extinction and metagenomics trials performed at 30ºC, and greater survival was observed at 15ºC in both soil treatments (Tables 2, 3).”

We agree that triplicate trials would improve the data, but the metagenomics trial was performed to assess bacterial taxa in each treatment and temperature over 28 days, not necessarily to determine E. coli O157 populations based on temperature.

Minor revisions

L.30. different

Response: The word “deferent” has been changed to “different”.

L93. Please, use a code to provide an identification at a strain level of the E. coli used in this work.

Response: Codes corresponding with previous research for each strain have been added.

L133-135. Provide more details about the analysis of organic compounds, macro- and micronutrients

Response: Details on each analysis for organic matter and macro-/micronutrients has been added as suggested by Reviewer 2 (see below).

“A subsample of autoclaved and natural soil was analyzed for organic matter content (Walkley-Black titration), pH, macro- and micronutrients (Mehlich-3 extraction and analysis via inductively coupled plasma atomic emission spectroscopy (ICP-AES)) at the UF IFAS Extension Soil Laboratory (Gainesville, FL).”

L142. 169 or 160?

Response: 169. This has been corrected throughout the text.

L147. Why the background microbiota in soils was carried out on days 0, 1, 3, 7, 14, 30, 56, and 84, and not only until 160 days as for E. coli determination?

Response: Background microbiota concentrations remained constant during the first 84 days, and once E. coli O157 populations were unrecoverable in natural soils (day 84), sampling was not performed.

L165. Mesocosm

Response: We have corrected the misspelling of “mescosm” to “mesocosm”

L240-243. Please, rephrase the sentence

Response: We have rephrased this sentence as suggested by Reviewer 2 along with the subsequent and related sentences for clarity.

Before: “Statistical analyses were not performed for the trial conducted at 15 and 30ºC in autoclaved and natural soils for metagenomic analysis – mesocosms were inoculated with separate E. coli O157 strains resulting in a lack of independent triplicate mesocosms for analysis. However, the mean log CFU g-1 (combined strains) was determined on each sampling day for comparison among soil treatments at each temperature (Table 3). Similar mean log CFU g-1 were observed between the trials performed at 30ºC, and greater survival was observed at 15ºC in both soil treatments (Tables 2, 3). On day 84, the mean log CFU g-1 at 15ºC was 4.81 ± 0.18 in autoclaved soils and 2.99 ± 0.74 in natural soils compared to being below the LOD in both experiments at 30ºC (Table 2, 3).”

After: “One trial was performed at 15 and 30ºC in autoclaved and natural soils for each E. coli O157 strain (metagenomics trial). The mean log CFU g-1 (combined strains) was determined on each sampling day for comparison among soil treatments at each temperature (Table 3). Statistical analysis was not performed based on mean log CFU g-1 in the metagenomics trail. However, similar mean log CFU g-1 were observed between the extinction and metagenomics trials performed at 30ºC, and greater survival was observed at 15ºC in both soil treatments (Tables 2, 3).”

L251-253. This information is provided in a confusing way

Response: We have updated this section as suggested by Reviewer 2 (see before and after below).

Before: “Although similar E. coli O157 concentrations were observed at 15 and 30ºC on day 28 in autoclaved soils at 6.91 ± 0.44 log CFU g-1 and 6.99 ± 0.11 log CFU g-1, respectively (Table 3), a greater than 5 log CFU g-1 decline in E. coli O157 was observed after 28 days at 30ºC while remaining at 4.50 ± 1.57 log CFU g-1 at 15ºC on day 140 (last sampling day).”

After: “Although similar E. coli O157 concentrations were observed at 15 and 30ºC from day 0 to 28 in autoclaved soils, E. coli O157 concentrations declined until extinction (day 140) at 30ºC while remaining at 4.50 ± 1.57 log CFU g-1 at 15ºC on day 140 (Table 3).”

L253-256. A similar difference (about 3-Log) is detectable at day 28

Response: We have added the values in natural soil on day 28 in addition to day 56 as suggested by Reviewer 2 (see below):

“ Similarly, greater survival was observed in natural soils at 15ºC versus 30ºC – on day 28, E. coli O157 concentrations were at 4.95 ± 0.54 at 15ºC versus 1.29 ± 1.03 at 30ºC. On day 56, E. coli O157 concentrations were at 3.52 ± 0.44 at 15ºC versus 0.70 ± 0.00 log CFU g-1 at 30ºC in natural soils (Table 3).”

L346-348. If no significant differences were found in the survival, the sentence is a no-sense

Response: As suggested by Reviewer 2, we have removed this sentence: “The stx1-/stx2- E. coli O157 strain used in this study survived longer in both autoclaved and natural soils in comparison to the stx1+/stx2+ and stx1-/stx2+ strains, although these differences in survival were not significant.”

However, we did add that “similar findings were observed in this study” when discussing virulence not impacting survival (next sentence)

L385-387. Rephrase the sentence in a more clear way

Response: We have rephrased this sentence as suggested by Reviewer 2.

Before: “Similar results were observed in the metagenomics trial, except the most drastic decline was observed between day 56 and 84 in the metagenomics trial versus day 30 and 56 in the extinction trials at 30ºC.”

After: “In the metagenomics trial, E. coli O157 declined to similar concentrations at each sampling point in comparison to the extinction trial at 30ºC. The most drastic decline in E. coli O157 was observed between day 56 and 84 in the metagenomics trial and between day 30 and 56 in the extinction trial at 30ºC.”

---

## [Decision Letter · Decision Letter 1]

29 May 2020

Survival of Escherichia coli O157 in Autoclaved and Natural Sandy Soil Mesocosms

PONE-D-20-03610R1

Dear Dr. Keith R. Schneider,

We are pleased to inform you that your manuscript has been judged scientifically suitable for publication and will be formally accepted for publication once it complies with all outstanding technical requirements.

With kind regards,

Luigimaria Borruso

Academic Editor

PLOS ONE

Additional Editor Comments (optional):

Reviewers' comments:

Reviewer's Responses to Questions

**Comments to the Author**

1. If the authors have adequately addressed your comments raised in a previous round of review and you feel that this manuscript is now acceptable for publication, you may indicate that here to bypass the “Comments to the Author” section, enter your conflict of interest statement in the “Confidential to Editor” section, and submit your "Accept" recommendation.

Reviewer #1: All comments have been addressed

Reviewer #2: All comments have been addressed

2. Is the manuscript technically sound, and do the data support the conclusions?

Reviewer #1: Yes

Reviewer #2: Yes

3. Has the statistical analysis been performed appropriately and rigorously? 

Reviewer #1: Yes

Reviewer #2: Yes

4. Have the authors made all data underlying the findings in their manuscript fully available?

Reviewer #1: Yes

Reviewer #2: Yes

5. Is the manuscript presented in an intelligible fashion and written in standard English?

Reviewer #1: Yes

Reviewer #2: Yes

6. Review Comments to the Author

Reviewer #1: (No Response)

Reviewer #2: The manuscript has been modified according to the suggestions of the reviewers, and it is now suitable for pubblication in PlosOne

7. PLOS authors have the option to publish the peer review history of their article (what does this mean?). If published, this will include your full peer review and any attached files.

Reviewer #1: No

Reviewer #2: No

---

## [Editor Report · Acceptance letter]

2 Jun 2020

PONE-D-20-03610R1 

Survival of *Escherichia coli* O157 in Autoclaved and Natural Sandy Soil Mesocosms 

Dear Dr. Schneider:

I'm pleased to inform you that your manuscript has been deemed suitable for publication in PLOS ONE. Congratulations! Your manuscript is now with our production department. 

Kind regards, 

on behalf of

Dr. Luigimaria Borruso 

Academic Editor

PLOS ONE